# d-Allulose Improves Endurance and Recovery from Exhaustion in Male C57BL/6J Mice

**DOI:** 10.3390/nu14030404

**Published:** 2022-01-18

**Authors:** Bingyang Liu, Yang Gou, Takamasa Tsuzuki, Takako Yamada, Tetsuo Iida, Sixian Wang, Ryoichi Banno, Yukiyasu Toyoda, Teruhiko Koike

**Affiliations:** 1Department of Sports Medicine, Graduate School of Medicine, Nagoya University, Nagoya 464-8601, Japan; byliu92@163.com (B.L.); gou.yang@i.mbox.nagoya-u.ac.jp (Y.G.); wangsixiansichuan@163.com (S.W.); ryouichi@med.nagoya-u.ac.jp (R.B.); 2Faculty of Pharmacy, Meijo University, Nagoya 468-8503, Japan; ttsuzuki@meijo-u.ac.jp (T.T.); toyuki@meijo-u.ac.jp (Y.T.); 3Research and Development, Matsutani Chemical Industry Co. Ltd., Itami 664-8508, Japan; takako-yamada@matsutani.co.jp (T.Y.); tetsuo-iida@matsutani.co.jp (T.I.); 4Research Center of Health, Physical Fitness and Sports, Nagoya University, Nagoya 464-8601, Japan

**Keywords:** d-allulose, aerobic performance, recovery, skeletal muscle, maximal aerobic speed, glycogen, blood lactate

## Abstract

d-Allulose, a rare sugar, improves glucose metabolism and has been proposed as a candidate calorie restriction mimetic. This study aimed to investigate the effects of d-allulose on aerobic performance and recovery from exhaustion and compared them with the effects of exercise training. Male C57BL/6J mice were subjected to exercise and allowed to run freely on a wheel. Aerobic performance was evaluated using a treadmill. Glucose metabolism was analyzed by an intraperitoneal glucose tolerance test (ipGTT). Skeletal muscle intracellular signaling was analyzed by Western blotting. Four weeks of daily oral administration of 3% d-allulose increased running distance and shortened recovery time as assessed by an endurance test. d-Allulose administration also increased the maximal aerobic speed (MAS), which was observed following treatment for >3 or 7 days. The improved performance was associated with lower blood lactate levels and increased liver glycogen levels. Although d-allulose did not change the overall glucose levels as determined by ipGTT, it decreased plasma insulin levels, indicating enhanced insulin sensitivity. Finally, d-allulose enhanced the phosphorylation of AMP-activated protein kinase and acetyl-CoA carboxylase and the expression of peroxisome proliferator-activated receptor γ coactivator 1α. Our results indicate that d-allulose administration enhances endurance ability, reduces fatigue, and improves insulin sensitivity similarly to exercise training. d-Allulose administration may be a potential treatment option to alleviate obesity and enhance aerobic exercise performance.

## 1. Introduction

The global epidemic of obesity and obesity-associated diseases, especially type 2 diabetes, is expected to continue to worsen [1]. Exercise and nutritional management are the most effective preventative and therapeutic measures for obesity and obesity-associated diseases. However, maintaining the recommended levels of physical activity and a suitable diet in our modern obesogenic society is difficult. Thus, there is a crucial need to develop effective exercise and dietary programs [2].

d-Allulose, also known as d-psicose, is a rare, functional sugar formed by the epimerization of d-fructose at the C-3 position [3,4]. d-Allulose has been shown to ameliorate insulin resistance [5,6,7] and glucose tolerance in rodents [8,9,10] and humans [11] and reduce abdominal fat accumulation in rodents [12,13,14] and humans [15,16].

ATP availability is essential for skeletal muscle contraction in sprint events (anaerobic) as well as endurance events (aerobic) [17]. Both fat and carbohydrates are major fuels for aerobic exercise [18]. Glycogen in muscles and the liver provides an energy source for aerobic performance [19], and glycogen accumulation can suppress body fatigue during exercise [20]. Alternatively, the efficient use of fat can contribute to exercise performance [21,22] and thus enhance endurance. We hypothesized that d-allulose would enhance exercise performance because its administration has been shown to increase muscle or liver glycogen storage and fat oxidation. Matsuo reported that daily intake of d-allulose may accelerate the repletion of liver and muscle glycogen after exhaustive swimming [23]. Long-term administration of a rare sugar syrup containing d-allulose enhanced the translocation of liver glucokinase and increased liver glycogen content [7]. In addition, Nagata et al. showed that d-allulose promoted fat oxidation [24], and enhancement of postprandial fat oxidation in response to d-allulose intake has also been reported in healthy humans [25]. To the best of our knowledge, the effects of d-allulose on aerobic performance have not been reported. Therefore, in the present study, we assessed whether d-allulose administration improves endurance ability and accelerates recovery from exhaustion. We further compared the effects of d-allulose with exercise training effects to understand its mechanism of action.

## 2. Materials and Methods

### 2.1. Animals and Diets

Six-week-old male C57BL/6J mice were purchased from SLC Co. Ltd. (Tokyo, Japan). The mice were singly housed in cages in a room maintained at 23 ± 2 °C under a 12 h light/12 h dark cycle and allowed free access to a standard chow diet (MF powders, Oriental Yeast Co. Ltd., Tokyo, Japan) and water from the period of environmental adaptation at week 1 until the beginning of the experiment. During the intervention period, mice were either kept on the chow diet (AIN93G powder, including 3% cellulose, purchased from Oriental Yeast Co. Ltd., Tokyo, Japan) or a d-allulose diet (AIN93G powder, including 3% d-allulose). The composition of the experimental diets is presented in Appendix A. d-Allulose was provided by Matsutani Chemical Industry Co. Ltd. (Hyogo, Japan). To standardize the number of calories in the diets, d-allulose was replaced with cellulose. A free-wheel running apparatus (Med Associates Inc., St Albans, VT, USA) was placed into individual cages housing mice assigned to exercise, and the mice were allowed to run freely on the wheels. Food intake was measured once every 2 days. Body weights were measured weekly. All experimental procedures involving animals followed the guidelines for the Care and Use of Laboratory Animals of Nagoya University. Ethical approval was granted by the Animal Experiment Committee of Nagoya University (HPFS No. 20).

### 2.2. Experimental Protocols

#### 2.2.1. Experiment 1: Effect of Long-Term d-Allulose Administration on Aerobic Performance

Following treadmill acclimatization (described later), 6-week-old mice underwent adaptation for 1 week and treadmill acclimatization for 3 weeks. Mice were randomly assigned into two groups: a chow diet group (E1 group, *n* = 6) and a d-allulose diet group (AE1 group, *n* = 7). Running wheels were placed in all cages. As shown in Figure 1, all mice performed the first endurance and recovery tests on a chow diet. The second endurance and recovery tests were performed after 4 weeks of administration of either a chow diet or a d-allulose diet. 

#### 2.2.2. Experiment 2: Effect of Long-Term d-Allulose Administration on Maximal Aerobic Speed and Physiological Indicators Related to Aerobic Performance

Ten-week-old mice were divided into four groups: sedentary/chow diet group (C2, *n* = 6), sedentary/d-allulose group (A2, *n* = 6), exercise/chow diet group (E2, *n* = 6), and exercise/d-allulose group (AE2, *n* = 7). To standardize initial endurance ability, the mice were grouped based on the lactate levels immediately after running on a treadmill for 2 h at 20 m/min (the first BGL test). Mice in the exercise groups (E2 and AE2) had free access to the wheel after the grouping at the start of week 3. As shown in Figure 2, the first MAS and BGL tests were performed before the grouping. The second tests were conducted 4 weeks after the grouping. Mice were euthanized by cervical dislocation immediately after running on the treadmill at 20 m/min for 30 min at the end of week 8. Muscles and liver were immediately removed and subsequently frozen in liquid nitrogen for further analysis.

#### 2.2.3. Experiment 3: Effect of Short-Term d-Allulose Administration on MAS, Blood Glucose, and Blood Lactate Levels

Ten-week-old mice (*n* = 6) were on an alternate feeding regimen of chow diet or d-allulose. We measured both glucose and lactate with a biosensor BF-5S instrument (Oji Scientific Instruments Co. Ltd., Hyogo, Japan) before and after each maximal aerobic speed (MAS) test, performed as shown in Figure 3.

### 2.3. Treadmill Acclimatization and Aerobic Performance Tests 

#### 2.3.1. Treadmill and Running Wheel Acclimatization 

Tests and training were performed on a motorized treadmill (10% grade; MK-690S, Muromachi Kikai Co. Ltd., Tokyo, Japan). Mice were first accustomed to the treadmill by running at 5 m/min for 10 min (warm-up), then speeding up to 10 m/min for 30 min, three times, and then the speed was increased to 20 m/min. Mice were encouraged to run by light electrical stimulation (0.2 mA) provided by a grid located at the rear end of the treadmill belt. The wireless running wheel apparatus (Med Associates Inc., St Albans, VT, USA) was used to monitor the movement of the mice. Rotation of the wheel by mice transmitted a wireless electronic signal to a hub, and the number of revolutions was recorded on Wheel Manager software (Med Associates Inc.) every few seconds. Activity was recorded as rotation over time, and the data were exported to a Microsoft Excel spreadsheet [26]. 

#### 2.3.2. MAS Test 

We followed the protocol by Pauly et al. [26], and conditions were optimized based on the settings of our treadmill machine. After a 10-minute warm-up at 5 m/min, the speed was increased by 2 m/min every 2 min. Upon reaching 20 m/min, the speed was increased by 1 m/min every 2 min until the mice could no longer maintain the pace. The MAS was the speed corresponding to the last stage completed by the mouse.

#### 2.3.3. Endurance Test 

Aerobic endurance was determined as the distance that a mouse could move when it reached a state of exhaustion at a moderate-intensity running speed of about 75% MAS [27]. After a 10 min warm-up at 5 m/min, the running speed was gradually increased to 20 m/min (average 75% MAS), which was then maintained. The test was terminated when the mice could no longer maintain the pace. Mice were judged to be in a state of exhaustion if they were shocked 20 times in 1 min [28].

#### 2.3.4. Recovery Test 

The recovery of mice from a maximum exercise load was evaluated by comparing the change in wheel running distance after the endurance tests. The recovery ratio was defined as the ratio of daily exercise volume for each mouse to the average daily running distance during the 7 days before the endurance test. The percentage of baseline activity [29] was used to adjust the baseline activity variability of mice to evaluate the voluntary exercise capacity. 

#### 2.3.5. Blood Glucose and Lactate Measurement before and after Running for 2 h at 20 m/min (BGL Test) 

Blood was obtained from the tail veins of mice. After a 10-minute warm-up at 5 m/min, the speed was increased by 2 m/min every 2 min until 20 m/min and then maintained for 2 h. Tail blood was collected and used to determine blood glucose and lactate levels before running (basal) and immediately after running (0 min).

### 2.4. ipGTT

Mice were subjected to fasting for 4 h before the test. The running wheels of the E2 and AE2 groups were removed 4 h before the ipGTT to avoid an acute effect of voluntary exercise. Each mouse received an intraperitoneal injection of glucose solution (2 g/kg BW), and blood samples were collected at 0, 15, 30, 60, and 120 min after the injection. Blood glucose levels were determined using the biosensor BF-5S instrument. Blood was also collected in Eppendorf tubes containing heparin (1:1000) at 15 min. Plasma was obtained following centrifugation (2000× *g* for 10 min at 4 °C) of blood, stored at −80 °C, and used to determine insulin levels with the mouse ELISA KIT (FUJIFILM Wako Sibayagi Corporation, Gunma, Japan) according to the manufacturer’s instructions.

### 2.5. Liver and Muscle Glycogen Measurement

The liver and soleus muscle glycogen levels were measured using a Glycogen Colorimetric Assay Kit II (BioVision, Milpitas, CA, USA) according to the manufacturer’s instructions.

### 2.6. Western Blotting 

Proteins in the soleus muscle were extracted using a homogenization buffer (50 mM HEPES, pH 7.4; 150 mM NaCl, 1.5 mM MgCl2, 0.01% trypsin inhibitor, 10% glycerol, 1% Triton X-100, 2 mM phenylmethylsulfonyl fluoride). The supernatants containing protein samples were obtained after centrifugation (7000× *g* for 30 min, 4 °C). We performed Western blotting using a previously described protocol [30]. Briefly, 40 or 100 µg of protein extracts was separated by SDS-PAGE at 20 mA and then transferred to polyvinylidene difluoride (PVDF) membranes (EMD Millipore Corporation, Billerica, MA, USA). After blocking the membranes with 5% non-fat milk in TBS-T buffer (20 mM Tris, 0.8% NaCl, 0.1% Tween 20) for 1 h at room temperature (25 °C), the membranes were incubated overnight at 4 °C with a 1:1500 dilution of primary antibodies against β-actin (13E5), anti-AMPK, anti-phospho-Thr172 AMPK, anti-ACC, anti-phospho-ACC (recognizes Ser79 of ACC1) (all from Cell Signaling Technology Inc., Danvers, MA, USA), or anti-PGC-1α (Proteintech Inc., Rosemont, IL, USA). Membranes were washed five times in TBS-T for 5 min each time, incubated in a 1:3000 dilution of horseradish peroxidase-conjugated goat anti-rabbit (Bio-Rad, Laboratories Inc., Hercules, CA, USA) or anti-mouse (KPL, Gaithersburg, MD, USA) IgG antibody for 1 h at room temperature, and washed five times in TBS-T. Immunoreactive bands were detected using an ECL detection system (GE Healthcare UK Limited, Buckinghamshire, UK). Images of membranes were obtained on film and analyzed using ImageJ software (National Institutes of Health, Bethesda, MD, USA). Individual data points for the control group were divided by the group mean. Thus, the mean of the normalized control group was 1 with variability. The density of protein bands for other groups was expressed as the fold change in density of the control group values.

### 2.7. Statistical Analyses

Results are shown as mean ± standard error of the mean (SEM). Data were analyzed using Student’s *t*-test and one-way ANOVA (using Tukey’s or Dunnett’s post hoc test) on Prism 8 software (GraphPad Software, San Diego, CA, USA). Differences at *p* < 0.05 were considered as being statistically significant.

## 3. Results

### 3.1. d-Allulose Improves Endurance and Recovery (Experiment 1)

In Experiment 1, all mice were allowed to run on the running wheel after acclimatization for 1 week. The voluntary running distance during the 4-week period after the grouping was higher in the d-allulose group than in the control group (Figure 4A). An increase in running distance in the d-allulose group was observed within ten days of the d-allulose administration (Appendix A).

In the endurance test, the average running distance of mice in the AE1 group was significantly higher than that of mice in the E1 group after 4 weeks of d-allulose administration (Figure 4B). The distances obtained in the second endurance test were significantly higher compared with those obtained in the first test for mice in both the E1 and AE1 groups. 

The recovery speed after the first endurance test was similar between the control and d-allulose groups before d-allulose administration (Figure 4C,D). However, its administration enhanced the recovery speed (Figure 4E,F). Of note, a significant difference was observed from the date of the endurance test (day 0). 

### 3.2. Effect of Long-Term Administration of d-Allulose, Exercise, and Their Combination (Experiment 2)

#### 3.2.1. d-Allulose Improves MAS

The MAS of mice in the C2 group decreased by approximately 10%, while that of mice in the other groups increased significantly (A2 group 12.7% (*p* < 0.05); E2 group 23.6% (*p* < 0.01); AE2 group 24.8% (*p* < 0.01); Figure 5A). The MAS of mice in the A2, E2, and AE2 groups after 4 weeks of treatment was significantly higher than that of mice in the C2 group (Figure 5B).

#### 3.2.2. d-Allulose Suppresses Blood Lactate Increase after Running

Blood glucose and lactate levels before and immediately after running for 2 h at 20 m/min are shown in Figure 6. Changes in the blood glucose and lactate levels before treatment were similar among the four groups (Figure 6A,B). Changes in the blood glucose level after treatment were also similar among the groups (Figure 6C). Of note, after treatment, the blood lactate levels immediately after running were significantly lower in the A2, E2, and EA2 groups compared with those in the C2 group (*p* < 0.001). 

#### 3.2.3. d-Allulose Improves Insulin Sensitivity

IpGTT profiles were similar among the four groups, except at 15 min (Figure 7A), and no significant differences in the AUC of blood glucose levels were found among the groups (Figure 7B). After administering glucose solution, the blood glucose levels peaked at 15 or 30 min. At 15 min, the blood glucose levels of mice in the AE2 group were significantly higher compared with those of mice in the other three groups (Figure 7C). Blood insulin levels at 15 min were lowest in mice in the AE2 group, and those of mice in the E2 and A2 groups were significantly lower than those of mice in the C2 group (Figure 7D). Taken together, the changes in blood insulin level indicated an improvement in systemic insulin sensitivity following the administration of d-allulose or exercise.

#### 3.2.4. d-Allulose Prevents Increase in Body Weight and White Adipose Tissue Weights

As shown in Table 1, the body weights of the mice were significantly higher in the C2 group compared with the other groups. Mice in the A2 group consumed approximately 20% less food than the mice in the C2 group. In contrast, mice in the E2 and AE2 groups consumed similar amounts of food to mice in the C2 group. The weight of adipose tissue was lower for mice in the A2 and E2 groups compared to that in the C2 group. The weight of adipose tissue of mice in the AE2 group was lower than that in the A2 and E2 groups. There were no significant differences in liver or muscle weights among the groups.

#### 3.2.5. d-Allulose Increases Liver Glycogen But Not Muscle Glycogen Levels

Liver glycogen levels were higher in mice in the A2 and E2 groups compared to mice in the C2 group, and the livers of mice in the AE2 group contained more glycogen than those of mice in the E2 group (Figure 8A). The difference in liver glycogen levels between the A2 and AE2 groups was statistically insignificant (*p* = 0.06). The muscle glycogen level increased in the E2 and AE2 groups but not in the A2 group (Figure 8B).

#### 3.2.6. Effect of d-Allulose on AMPK, ACC, and PGC-1α in Skeletal Muscle

Western blotting of protein samples isolated from soleus muscle removed immediately after running on the treadmill for 30 min at 20 m/min (Figure 9A) revealed enhanced AMPK phosphorylation in the A2, E2, and AE2 groups compared with the C2 group (Figure 9B). The total amount of AMPKα was similar among samples from the four groups (Figure 9C). The expression of PGC-1α was higher in samples from the A2, E2, and AE2 groups compared with those from the C2 group (Figure 9D). Phosphorylation of ACC was also higher in samples from the A2, E2, and AE2 groups compared with those from the C2 group (Figure 9E). The expression of ACC was lower in samples from the A2 group compared with those from the C2 group and was lower in samples from the E2 and AE2 groups compared with those from the A2 group (Figure 9F).

### 3.3. Effect of Short-Term d-Allulose Administration on MAS, Blood Glucose, and Blood Lactate Levels (Experiment 3)

We conducted a crossover study to examine the effects of short-term d-allulose administration on MAS because in Experiment 2, with 4 weeks of administration of d-allulose, differences in body weights and fats were observed among mice from the groups, which might have affected the MAS results. In this experiment, the MAS increased even after short-term administration of d-allulose and decreased during the period of chow diet administration (Figure 10A). Blood lactate levels immediately after running were significantly decreased following the administration of d-allulose, despite the increased MAS (Figure 10B). Blood glucose levels decreased during the period of chow diet administration but not during the d-allulose administration period (Figure 10C).

## 4. Discussion

In the present study, we investigated the effect of d-allulose on the exercise performance of C57BL/6J mice. We, and others, have shown that d-allulose administration improves glucose metabolism and insulin resistance in diet (high-sucrose or high-fat)-induced obese rodents, db/db mice, type 2 diabetes model rats, and humans with borderline diabetes [5,6,7,8,9,10,11]. To our knowledge, the effects of d-allulose on aerobic exercise performance have not been previously reported. Interestingly, d-allulose administration increased running distance and running speed and induced a rapid recovery from exhaustive running. Furthermore, d-allulose administration improved insulin sensitivity and induced changes in signaling molecules that were similar to those observed with exercise.

The effects of d-allulose on aerobic exercise performance can depend on the availability of carbohydrates and fat, which are the main sources for ATP production. Glycogen contents in the muscles and liver are associated with endurance and recovery from exhaustive exercise. Although muscle glycogen is essential for ATP production during exercise in humans [19], the contribution of liver glycogen may be more significant in mice [31]. The proportion of liver glycogen in rodents is 10-fold higher than that in humans [32]. Pederson et al. reported that mouse muscle glycogen is not essential during strenuous exercise because genetic depletion of muscle glycogen did not affect the endurance ability of mice [33]. Recently, Lopez-Soldado et al. reported that increased liver glycogen levels enhance aerobic capacity in mice [34]. In our study, both the voluntary running distance and the MAS increased after short-term administration of d-allulose. These effects of d-allulose may be due to increased levels of liver glycogen. However, d-allulose can also impact exercise performance through changes in fat oxidation.

Lipids are another primary source for the production of ATP. During muscle contraction, the uptake of fatty acids in skeletal muscle increases, while the activity of ACC decreases, which promotes the phosphorylation of ACC [35]. PGC-1α is involved in the regulation of energy metabolism and mitochondrial biogenesis [36]. Long-term aerobic exercise training significantly affects the AMPK-PGC-1α pathway [37]. We found that d-allulose administration enhanced the AMPK axis, which is consistent with d-allulose enhancing the oxidation of fatty acids [24,25]. A strong fat-oxidation ability increases the endurance exercise capacity [22]. An increase in fat oxidation following d-allulose administration has been demonstrated in rats [24] and humans [25], which can explain the improved exercise performance observed in the present study. 

The accumulation of blood lactate indicates the aerobic/anaerobic transition and is an important marker for endurance exercise capacity [38]. Lactate has been reported to be the primary cause of muscle fatigue; however, recent evidence suggests that lactate level does not correlate with muscle fatigue [39]. In the present study, blood lactate levels after 2 h of endurance running at moderate intensity were lower after 4 weeks of d-allulose administration or exercise training. As the intensity of endurance running was likely to be below the lactate threshold levels [21], the difference in lactate levels is possibly because of the improved utilization of fatty acid rather than the increased synthesis of lactate. In the MAS test, even short-term administration (e.g., 3 days) of d-allulose effectively reduced lactate levels after running. As lactate levels are expected to increase if increased liver glycogen is driving the improvement in MAS, we speculate that the improvement in MAS can be, at least in part, because of improved fat oxidation. Blood glucose levels significantly decreased during the period of chow diet administration but not during the d-allulose administration period, which suggests that d-allulose administration increased glycogen levels or the preferred use of free fatty acid. 

Mitochondrial changes accompanied improvements in fat oxidation. Activation of AMPK-PGC-1α signaling induces mitochondrial synthesis. It remains unclear whether these adaptations in muscle metabolism occur within a few days of d-allulose administration. Metabolic adaptation (i.e., lower exercise glycogen loss and lactate concentration during exercise after training) has been reported to occur even before mitochondrial changes [40,41]. Similar to exercise training, d-allulose may exert acute and chronic effects on exercise performance. The acute effect may be associated with its effect on glycogen levels. A previous report [23] and our results on MAS indicate that one or a few days of d-allulose administration can alter glycogen levels in muscle or the liver and aerobic performance. In contrast, muscle adaptation induced by d-allulose administration, accompanied by changes in the expression of AMPK, ACC, and PGC-1α, takes time. Studies have focused on the role of AMPK as a mediator of cell signaling pathways related to skeletal muscle function and metabolism [37].

d-Allulose reduces visceral fat and body weight and improves insulin sensitivity. Iwasaki et al. [42] reported that d-allulose administration reduced food intake mediated by GLP-1 release and vagal afferent activation. d-Allulose administration also increases energy consumption [43]. We previously showed that d-allulose administration reversed insulin resistance induced by a high-sucrose [5] and high-fat [6] diet. In the present study, body weights were reduced by d-allulose administration in both experiments, which reduced food intake in the high-sucrose diet group but not in the high-fat diet group. In the present investigation, d-allulose administration reduced both food intake and body weight. 

Interestingly, d-allulose administration did not affect food intake and body weight in the exercise groups. Further research is needed to determine how exercise is associated with the effects of d-allulose on food intake or energy expenditure. The ipGTT results indicated that d-allulose administration or exercise enhances insulin sensitivity, suggesting that exercise and d-allulose affect insulin sensitivity through different mechanisms.

Owing to its chronic effect on body weight and insulin resistance, d-allulose has been proposed as a caloric restriction mimetic [44,45,46]. We suggest d-allulose as an exercise mimetic, which shares common features with caloric restriction mimetics. The present data indicated that d-allulose can prevent fatigue during or after exercise. The limitation of our study is the lack of direct evidence that improved FFA utilization contributes to the changes in aerobic performance. We also used only male mice, and sex-specific differences in the effect of d-allulose were not tested. To draw conclusions on the effect of d-allulose on insulin sensitivity in normal mice, an additional study using a hyperinsulinemic–euglycemic clamp method is necessary. Further research with human subjects is warranted to explore the anti-fatigue effects of d-allulose. Nevertheless, our study not only provides valuable insights into the potential role of d-allulose in alleviating obesity and enhancing aerobic exercise performance in humans but also conclusively establishes a viable and promising option to ameliorate the clinical outcomes of obesity-induced health problems by using d-allulose.

## Figures and Tables

**Figure 1 nutrients-14-00404-f001:**
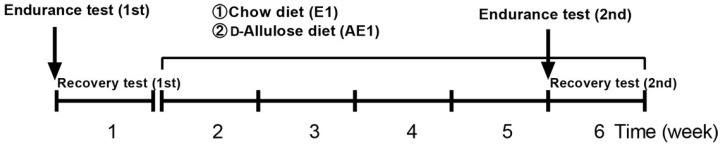
Experimental design of Experiment 1. The treatment period was from week 2 to week 6.

**Figure 2 nutrients-14-00404-f002:**
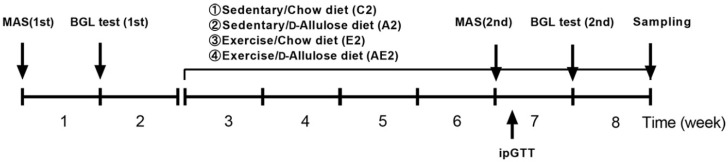
Experimental design of Experiment 2. The treatment period was from week 3 to week 8. MAS: maximal aerobic speed; BGL: blood glucose and lactate; ipGTT: intraperitoneal glucose tolerance test.

**Figure 3 nutrients-14-00404-f003:**
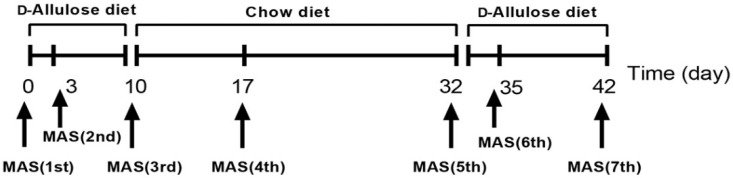
Experimental design of Experiment 3. The treatment periods were from day 0 to day 10 and day 32 to day 42.

**Figure 4 nutrients-14-00404-f004:**
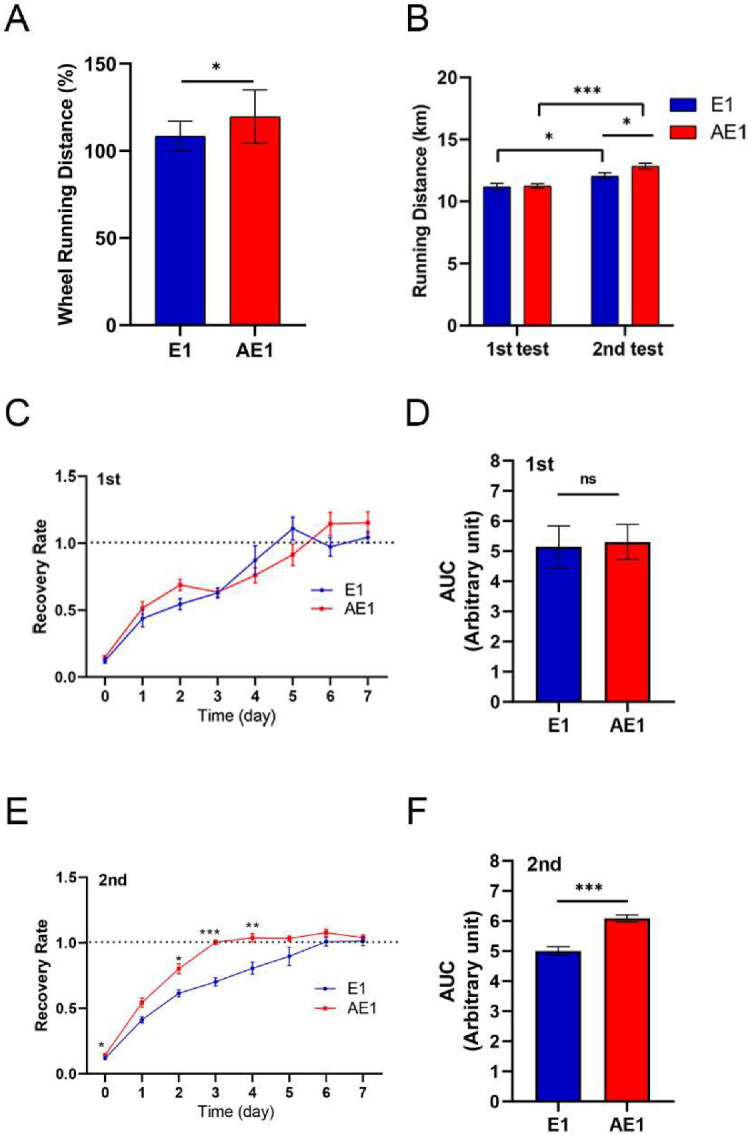
Effect of long-term d-allulose administration on endurance and recovery (Experiment 1). (**A**) Voluntary running distance during 4 weeks of d-allulose administration. The average running distance was calculated as a percentage of baseline activity, the average distance of the 3 days before the grouping. Changes in distance during the 4-week administration period are presented in Appendix A. (**B**) Running distance during the endurance tests before (first test) and 4 weeks after (second test) d-allulose administration. Daily recovery rate after the first (**C**) and second (**E**) endurance tests and the AUC of **C** (**D**) and **E** (**F**) (days 0–7). The daily recovery rate was calculated as a percentage of baseline activity, the average distance of 7 days before each endurance test. Data are shown as the mean ± SEM. Differences in recovery rate between E1 and AE1 groups (**C**,**E**) were analyzed using a repeated-measures *t*-test. *n* = 6 (E1 group) and 7 (AE1 group). * *p* < 0.05, ** *p* < 0.01, and *** *p* < 0.001. AUC: area under the curve; E1: exercise group fed with chow diet; AE1: exercise group fed with d-allulose diet.

**Figure 5 nutrients-14-00404-f005:**
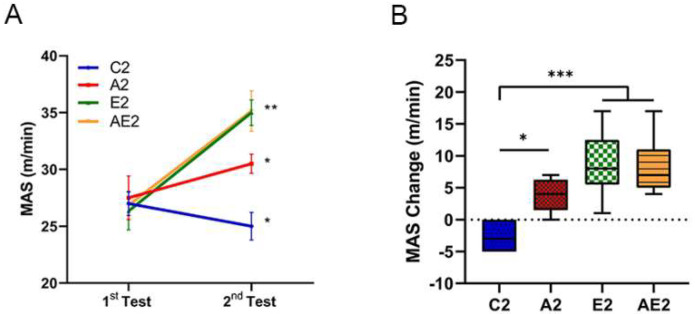
Effect of long-term d-allulose administration on MAS (Experiment 2). (**A**) MAS. Differences between the first and second tests were determined using a paired *t*-test. (**B**) Change in MAS between the first and second tests. Differences were analyzed using one-way ANOVA with post hoc Tukey’s test. *n* = 6 (C2, A2, E2 groups) and 7 (AE2 group). * *p* < 0.05, ** *p* < 0.01, and *** *p* < 0.001. Data are shown as the mean ± SEM. C2: sedentary group fed with chow diet; A2: sedentary group fed with d-allulose diet; E2: exercise group fed with chow diet; AE2: exercise group fed with d-allulose diet.

**Figure 6 nutrients-14-00404-f006:**
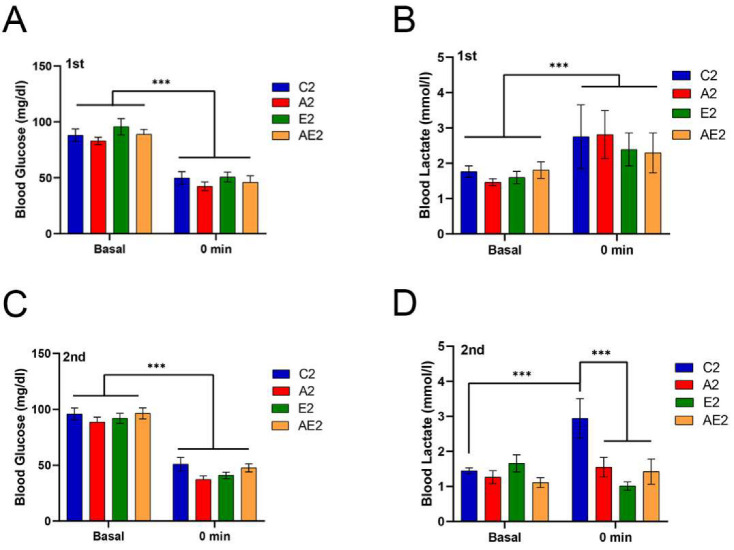
Effect of long-term d-allulose administration on blood glucose and lactate levels (Experiment 2). (**A**) Blood glucose and (**B**) blood lactate levels before running (basal) and immediately after (0 min) running in the first blood glucose and lactate measurement (BGL) test. (**C**) Blood glucose and (**D**) blood lactate levels in the second test. Data are shown as the mean ± SEM. Differences were analyzed using a *t*-test. *n* = 6 (C2, A2, E2 groups) and 7 (AE2 group). *** *p* < 0.001. C2: sedentary group fed with chow diet; A2: sedentary group fed with d-allulose diet; E2: exercise group fed with chow diet; AE2: exercise group fed with d-allulose diet.

**Figure 7 nutrients-14-00404-f007:**
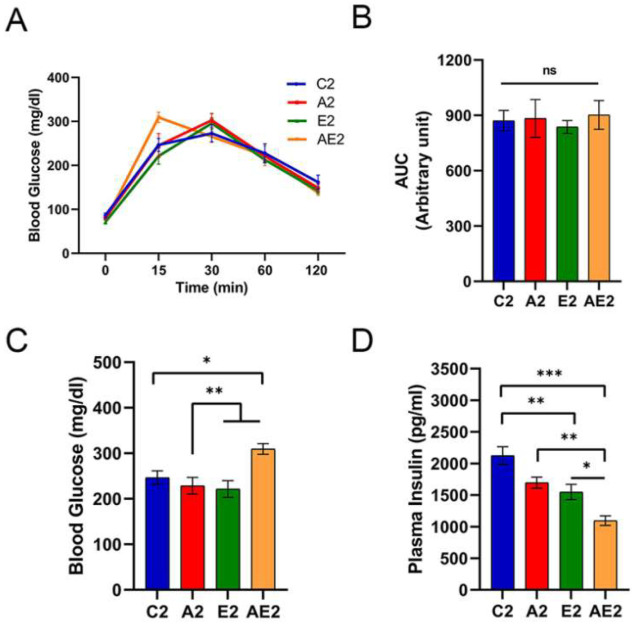
Effect of long-term d-allulose administration on ipGTT (Experiment 2). (**A**) Blood glucose levels during ipGTT, (**B**) AUC during ipGTT, and (**C**) blood glucose and (**D**) plasma insulin levels at 15 min after intraperitoneal injection. Data are shown as the mean ± SEM. Differences were analyzed using one-way ANOVA with post hoc Tukey’s test. *n* = 6 (C2, A2, E2 groups) and 7 (AE2 group). * *p* < 0.05, ** *p* < 0.01, and *** *p* < 0.001. AUC: area under the curve; C2: sedentary group fed with chow diet; A2: sedentary group fed with d-allulose diet; E2: exercise group fed with chow diet; AE2: exercise group fed with d-allulose diet.

**Figure 8 nutrients-14-00404-f008:**
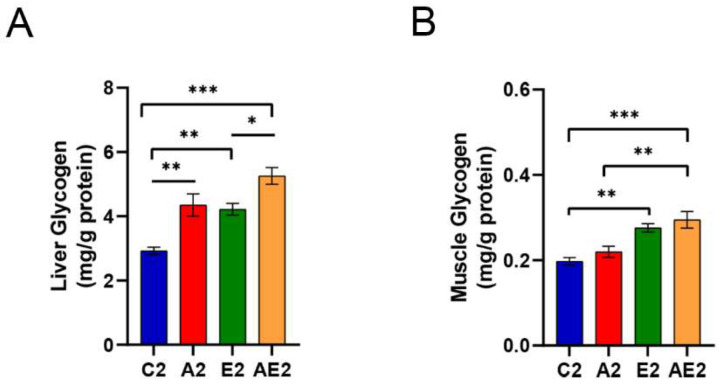
Liver and muscle glycogen levels following long-term d-allulose administration (Experiment 2). (**A**) Liver and (**B**) muscle glycogen levels. Data are shown as the mean ± SEM. Differences were evaluated using one-way ANOVA with post hoc Tukey’s test. *n* = 6 (C2, A2, E2 groups) and 7 (AE2 group). * *p* < 0.05, ** *p* < 0.01, and *** *p* < 0.001. C2: sedentary group fed with chow diet; A2: sedentary group fed with d-allulose diet; E2: exercise group fed with chow diet; AE2: exercise group fed with d-allulose diet.

**Figure 9 nutrients-14-00404-f009:**
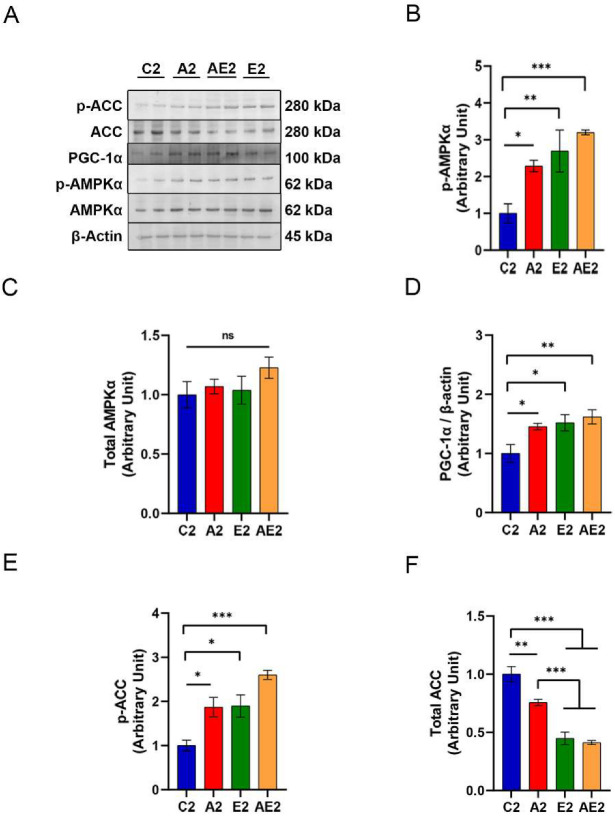
Effect of long-term d-allulose administration on AMPK and downstream signaling molecules in soleus muscles (Experiment 2). (**A**) Representative Western blot. (**B**) Phosphorylation at Thr172 and (**C**) expression of AMPK, (**D**) PGC-1α expression, (**E**) phosphorylation of ACC at Ser79, and (**F**) expression of ACC. Data are shown as the mean ± SEM. Differences were evaluated using one-way ANOVA with post hoc Dunnett’s (**B**–**E**) or Tukey’s (**F**) test. *n* = 6 (C2, A2, E2, AE2). * *p* < 0.05, ** *p* < 0.01, and *** *p* < 0.001. Uncropped images of the Western blots used in this analysis are shown in Appendix A. AMPK: AMP-activated protein kinase; p-AMPK: phosphorylated AMPK; ACC: acetyl-CoA carboxylase; p-ACC: phosphorylated ACC; PGC-1α: peroxisome proliferator-activated receptor γ coactivator 1α. C2: sedentary group fed with chow diet; A2: sedentary group fed with d-allulose diet; E2: exercise group fed with chow diet; AE2: exercise group fed with d-allulose diet.

**Figure 10 nutrients-14-00404-f010:**
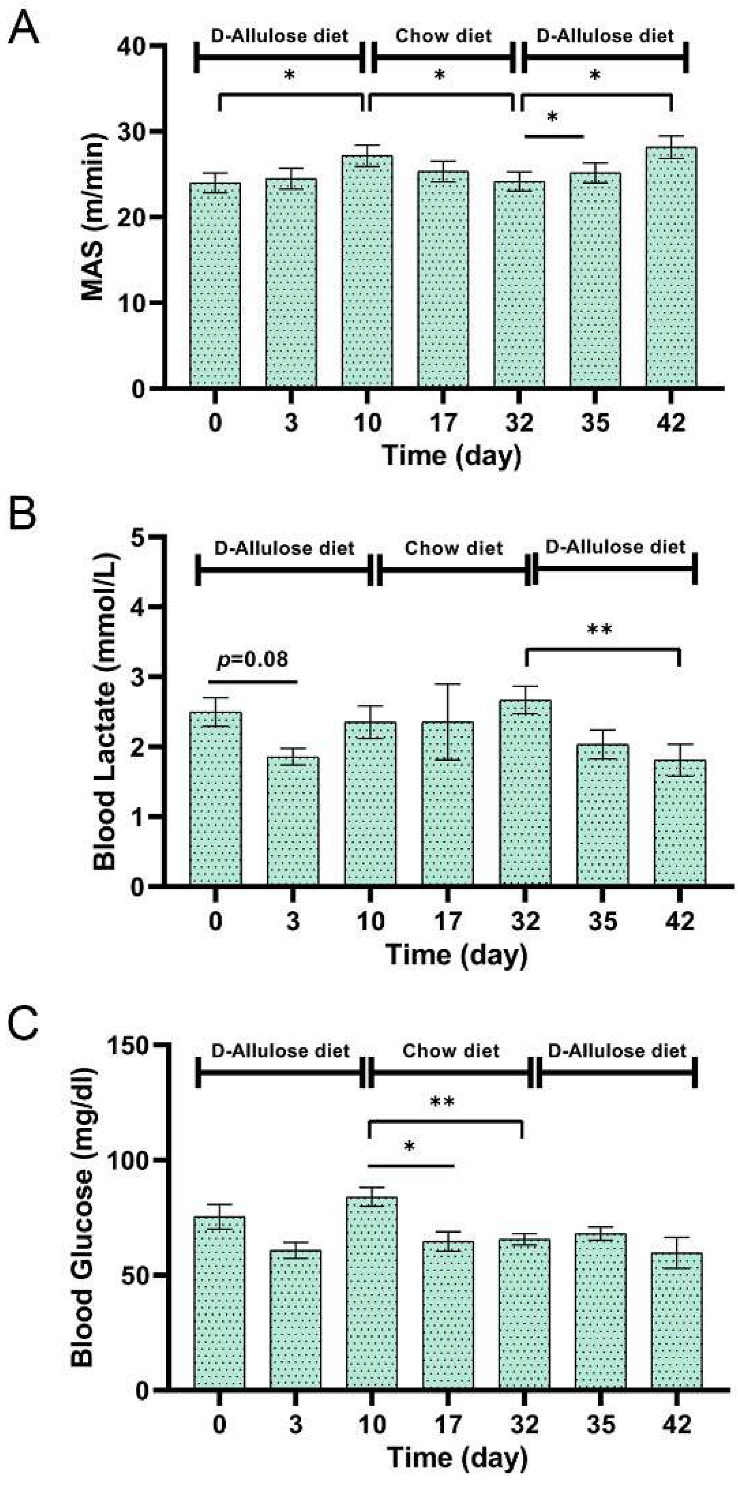
Effect of short-term d-allulose administration on MAS, blood glucose, and blood lactate levels (Experiment 3). (**A**) MAS, (**B**) blood lactate, and (**C**) blood glucose after MAS test. Data are shown as the mean ± SEM. Differences were analyzed using a paired *t*-test. *n* = 6. * *p* < 0.05 and ** *p* < 0.01. MAS: maximal aerobic speed.

**Table 1 nutrients-14-00404-t001:** Growth parameters.

Growth Parameters	C2 (*n* = 6)	A2 (*n* = 6)	E2 (*n* = 6)	AE2 (*n* = 7)
Initial body weight (g)	26.0 ± 0.4	25.6 ± 0.3	26.0 ± 0.2	25.9 ± 0.4
Final body weight (g)	30.4 ± 0.9	27.5 ± 0.4 ^#^	27.5 ± 0.5 ^#^	26.8 ± 0.4 ^#^
Food consumption (g/day)	4.7 ± 0.2	3.8 ± 0.2 ^#^	4.3 ± 0.2	4.1 ± 0.2
Liver (g)	1.114 ± 0.038	1.203 ± 0.024	1.075 ± 0.059	1.070 ± 0.014
Gastrocnemius (g)	0.242 ± 0.015	0.253 ± 0.006	0.285 ± 0.014	0.253 ± 0.006
Plantaris (g)	0.031 ± 0.002	0.039 ± 0.004	0.033 ± 0.003	0.034 ± 0.001
Soleus (g)	0.021 ± 0.004	0.030 ± 0.009	0.025 ± 0.006	0.016 ± 0.001
Tibialis anterior (g)	0.089 ± 0.003	0.112 ± 0.020	0.112 ± 0.013	0.097 ± 0.021
Extensor digitorum longus (g)	0.063 ± 0.013	0.045 ± 0.005	0.051 ± 0.011	0.045 ± 0.003
Epididymal fat (g)	1.087 ± 0.051	0.656 ± 0.035 ^#,^ *	0.512 ± 0.034 ^#,^ *	0.300 ± 0.032 ^#^
Perirenal fat (g)	0.357 ± 0.016	0.102 ± 0.017 ^#,^ *	0.106 ± 0.023 ^#,^ *	0.033 ± 0.006 ^#^
Mesenteric fat (g)	0.600 ± 0.013	0.341 ± 0.012 ^#,^ *	0.365 ± 0.026 ^#,^ *	0.237 ± 0.039 ^#^

Data are shown as the mean ± SEM and were analyzed by one-way ANOVA. ^#^: A2, E2, and AE2 vs. C2; *: A2 and E2 vs. AE2. *p* < 0.05. Exercise groups with free access to the wheel. C2: sedentary group fed with chow diet; A2: sedentary group fed with d-allulose diet; E2: exercise group fed with chow diet; AE2: exercise group fed with d-allulose diet.

## Data Availability

The data presented here are available on request from the corresponding author.

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
