# Peer review of "d-Allulose Improves Endurance and Recovery from Exhaustion in Male C57BL/6J Mice"

_nutrients, 2022, doi:10.3390/nu14030404_

Round 1

Reviewer 1 Report

Please, consult attached file.

Reviewer 2 Report

The authors have attempted to investigate the impact of D-Allulose dosing upon endurance and recovery from exercise in mice. The study is timely and important. Three experiments were conducted with different age mice and different lengths of treatment. 

There are some flaws in the presentation (quite confusing to remember which group was treated for how long and when the samples were taken)    and interpretation of the data (the lactate levels do not conform to your interpretation and insulin sensitivity did not happen) - the red numbered issues below. 

In Materials and methods-

1) around line 76 -are the animals singly housed? Required for voluntary running wheels and feed weighing, but not stated.

2) line 107 - Are you describing the 8 week sampling? Please make it more clear.

3) Line 115 - How often were blood glucose and lactate levels taken?

4) Line 155 - How were lactate levels measured?

5) Line 206 - remind the readers that the first test is basal, before any treatments or exercise, and that the second test was at 4 weeks of Rx. 

Results

6) Line 226 and 227 - ...the recovery speed after the second endurance test (Figu....date of the first endurance test... The reader should not be required to continually flip back and forth to figures 1, 2 and 3. Make it easy for them to read, even if they just read the figures.

7) Line 243 - I do not see a reduction of lactate levels after running. In graph 6D AE2 is larger than E2?

8) The figure legend needs to add the length of Rx for each mouse group

9) Line 258 - I also do not think that D-Allulose improves insulin sensitivity. If it did then the blood glucose would not increase in Figure 7C for AE2. Graph 7D shows less insulin is made, but this says nothing about muscle insulin sensitivity.

9) Table 1 is not complete in my version. I have 2 values for EDL weight and no values for the fat weights.

10) Figure 9 - what is the N= for the blot quantitations? How did you compare between gels? 

11) Figure 10 - graphs B and C are swapped.

12) Line 332 - ...administration on MAS. Because in experiment 2, with...

Round 2

Reviewer 2 Report

Thank you for the improvements to the manuscript.

This manuscript is a resubmission of an earlier submission. The following is a list of the peer review reports and author responses from that submission.